# Human Salivary Microbiota Diversity According to Ethnicity, Sex, TRPV1 Variants and Sensitivity to Capsaicin

**DOI:** 10.3390/ijms252111585

**Published:** 2024-10-29

**Authors:** Elena Vinerbi, Gabriella Morini, Claudia Picozzi, Sergio Tofanelli

**Affiliations:** 1Dipartimento di Biologia, Università di Pisa, 56126 Pisa, Italy; elena.vinerbi@irgb.cnr.it; 2Institute of Genetic and Biomedical Research (IRGB), National Research Council (CNR), 09042 Monserrato, Italy; 3Università di Scienze Gastronomiche, 12060 Pollenzo, Italy; g.morini@unisg.it; 4Dipartimento di Scienze per gli Alimenti, la Nutrizione e l’Ambiente, Università Degli Studi di Milano, 20133 Milano, Italy; claudia.picozzi@unimi.it

**Keywords:** Human microbiota, salivary microbiota, ethnicity, sex, *TRPV1* gene, capsaicin

## Abstract

The salivary microbiota of Italian and sub-Saharan African individuals was investigated using Nanopore sequencing technology (ONT: Oxford Nanopore Technologies). We detected variations in community composition in relation to endogenous (ethnicity, sex, and diplotypic variants of the *TRPV1* gene) and exogenous (sensitivity to capsaicin) factors. The results showed that *Prevotella*, *Haemophilus*, *Neisseria*, *Streptococcus*, *Veillonella*, and *Rothia* are the most abundant genera, in accordance with the literature. However, alpha diversity and frequency spectra differed significantly between DNA pools. The microbiota in African, male *TRPV1* bb/ab diplotype and capsaicin low-sensitive DNA pools was more diverse than Italian, female *TRPV1* aa diplotype and capsaicin high-sensitive DNA pools. Relative abundance differed at the phylum, genus, and species level.

## 1. Introduction

The microbial communities resident at different sites of various anatomical locations in the body, indicated as the microbiota, have been proven to have a role in its host’s physiological functions such as metabolism, immune development, and behavioral responses and therefore in health and diseases [1]. The Human Microbiome Project (HMP) [2] has recognized the presence of a central microbiome (core) and a variable microbiome. The first represents the genomic complex of taxa equal for each habitat and individuals, which constitutes a very small portion of the totality, about 5–10%. The second is the genomic complex of taxa that changes between body districts and individuals in relation to factors such as host genotype, lifestyle, and physiological state. The oral habitats, compared to other body sites, show higher alpha diversity (diversity within a sample) but lower beta diversity (diversity between samples). This indicates that the oral site shows more heterogeneity in terms of microbiome than other body habitats, being second only to the gut [3,4].

The six most abundant phyla are Firmicutes, Bacteroidetes, Proteobacteria, Actinobacteria, Fusobacteria, and Spirochaetes. These account for about 96% of the total taxa found in the samples analyzed so far. The most common genera are *Streptococcus*, *Eubacterium*, *Selenomonas*, *Veillonella* (Firmicutes), *Actinomyces*, *Atopobium*, *Rothia* (Actinobacteria), *Neisseria*, *Eikenella*, and *Campylobacter* (Proteobacteria), *Fusctobaeria*, and *Leptotrichia*, (Fusobacteria), *Prevotella*, *Capnocytophaga*, and *Porphyromonas* (Bacteriodetes) [3,4,5,6].

The relative abundance of the detected microorganisms in the oral cavity is affected by the sampled sites, saliva being one of them [3]. The salivary microbiota has attracted attention in the last decade and has been proven its stability over a reasonable period [7], making it a reliable biomarker. Saliva collection, storage, and DNA extraction methods have also been studied and compared, indicating that they do not have a major influence on microbiome profiles [8], thus confirming saliva as a reliable and significant sample of the human oral microbiota.

The salivary microbiota is affected by endogenous factors such as age, sex, ethnicity, and host genotype [9,10,11,12,13]. Therefore, can undergo significant and reversible changes due to the influence of external factors such as diet, smoking, and intake of antibiotics or alcohol [9,14,15,16,17,18,19]. Oral microbiota variability factors are also confirmed from other sample sites, and, among others, we refer to some papers worth reading for the relevance of the considered sample [20,21] or the kind of study (longitudinal) [22].

While oral microbiota appears to modulate taste perception of the host through metabolite production, influencing dietary preferences, some researchers have identified a relationship between taste receptor genetics and oral microbiota composition [23,24], due to the capability of some bitter taste receptors to respond to bacterial molecules by activating innate immunity [25,26,27]. Transient receptor potential (TRP) channels, involved in physical and chemical stimuli sensing and able to respond to temperature, pH, osmolarity, pheromones, and plant compounds [28,29], have also been proven to have a role as sensors of bacterial endotoxins and quorum sensing molecules [30,31]. But while their role in shaping the gut microbiota has been broadly investigated (for an extensive review see [32]), very little has been done regarding their contribution to oral microbiota composition [33].

Capsaicin, a vanillylamide, is the main secondary metabolite conferring the chemesthetic property of hotness to plants of the genus *Capsicum* through the activation of TRPV1 (transient receptor potential vanilloid 1) [34]. The other receptors responsible for chemesthetic sensations in the oral cavity and the nose are TRPM8 (melastatin, coolness) and TRPA1 (ankyrin, pungency) [29]. To our knowledge, capsaicin has been extensively studied for its role in the composition of the gut microbiota [35,36,37], not in the composition of the oral one.

A paper was recently published on the effect of mint oils on the composition of the human oral microbiome, but the possible contribution of TRPM8 has not been investigated [38].

The TRPV1 activation threshold temperature (around 42 °C in humans) is lowered by vanilloids and several other natural compounds, such as piperine in pepper and gingerols in ginger. pH values below 6, a level easily reached by tissue injury due to infection and inflammation, are also able to activate it [39], conferring to TRPV1 a role in the process of injury-related hyperalgesia, inflammation, and pain [40]. Two haplotype blocks in the *TRPV1* gene (H1 and H2) showed haplotypes with a variability pattern compatible with a stabilizing selection model (frequencies around 50%, significantly positive Tajima’s D values), only in individuals with sub-Saharan African origin. The two regions have been used to investigate the correlation with body composition and sensitivity to capsaicin [41]. Significative differences were observed for body composition parameters but not for capsaicin sensitivity. African women carrying the H1-b and H2-b haplotypes showed lower extracellular fluid retention and a higher percentage of fat mass, whereas no significant association was found in men.

In this study, we used Nanopore sequencing technology (ONT: Oxford Nanopore Technologies) to sequence the metagenome of salivary microbiota in male and female volunteers of Italian and sub-Saharan African ancestry. We aimed at investigating whether endogenous (ethnicity, sex, *TRPV1* diplotypic variants) and exogenous (sensitivity to capsaicin) factors are related to changes in the oral microbiota.

## 2. Results

### 2.1. Quality Control

The total reads obtained were 651,199. From these, we filtered out the reads classified as *Homo sapiens* (577,759), as root (8738), those unclassified (54,520), those showing classification inconsistency (55), and those with single counts (N = 11,883). Overall, the reads that passed the filtering were 20,924 (Figure 1).

### 2.2. Comparisons Between Human Salivary Microbiota Communities

Analytical results by taxonomic levels are summarized in Appendix A.

#### 2.2.1. Ethnic Differences

The comparison by ethnicity was performed by comparing the salivary DNA pool (pDNA) of sub-Saharan Africans (AFR, reads = 7321) with the pDNA of Italians (ITA, reads = 2931). 

At the phylum level (Appendix A), the AFR pool showed a more diverse salivary microbiota (Shannon diversity index H: AFR = 1.61 vs. ITA = 1.41, t = 9.73; *p* << 0.05). Six phyla—Proteobacteria, Bacteroidetes, Firmicutes, Actinobacteria, Fusobacteria, and Spirochaetes—occupied almost 96% of the total in both groups, in accordance with the literature [3]. Also, at the genus level (Figure 2, Appendix A), the frequency was significantly different (t = 13.46; *p* <<< 0.05), with the AFR pool showing a more diverse (Shannon H: AFR = 2.89, ITA = 2.41) microbiota. An increase between 4 and 6% in the genera *Rothia*, *Veillonella*, and *Streptococcus* was observed in the AFR pool. A similar increase for *Neisseria* (5.4%) and a much more marked increase for *Prevotella* (18.6%) was observed in the ITA pool. At the species level (Appendix A), the differences mirrored those observed at the level of the upper taxonomic ranks (Shannon H: ITA = 3.43, AFR = 3.87; t = 11.32, *p* << 0.05). Interestingly, the contribution of two species of the genus *Haemophylus* (*H. parahaemolyticus* + 9.4% in the ITA pool and *H. parainfluenza* + 9.0% in the AFR pool) was not detectable at the genus level.

#### 2.2.2. Differences Between Sexes

The comparison by sex was performed by comparing the pDNA of the males (M, reads = 4467) with the pDNA of the females (F, reads = 5785). 

At the phylum level (Appendix A), the diversity between the two pools was significant (Shannon H: t = 3.11; *p* < 0.05). The M pool has a more diverse salivary microbiota (H = 1.61) than pool F (H = 1.55). The more abundant phyla were Bacteroidetes in the M pool, Firmicutes, and Actinobacteria in the F pool. At the genus level as well (Figure 3, Appendix A), the difference between diversity indices was significant (t = 13.71; *p* <<< 0.05), with the M pool (H = 2.97) showing a more diverse microbiota composition than the female pool (H = 2.59). An increase between 4 and 6% in the genera *Rothia*, *Veillonella*, and *Neisseria* was observed in the F pool, while an increase of the genus *Porphyromonas* (3.4%) was observed in the M pool. At the species level (Appendix A), the differences reflected those of the higher taxonomic rank levels (Shannon H: M = 4.01, F = 3.66; t = 11.60, *p* << 0.05).

We also investigated whether sex differences would affect ethnic differences. When the different pDNAs were disaggregated (Appendix A), only comparisons between ITA/AFR males showed a significant difference at the genus level (Shannon H: ITA-M = 2.18, AFR-M = 3.07; t = 13.91, *p* <<< 0.05), not between females (Shannon H: ITA-F = 2.44, AFR-F = 2.52; t = 1.82, *p* > 0.05). This suggests that the most diverse oral microbial community of African males may affect both sex and ethnic differences.

#### 2.2.3. Differences Between *TRPV1* Diplotypes

The comparison by *TRPV1* H1 diplotypes was performed only on sub-Saharan Africans, subdivided into individuals bearing diplotypes aa (AploA, reads = 4290) and individuals bearing diplotypes bb/ab (AploB, reads = 3031).

At the phylum level (Appendix A), the diversity (Shannon H) between the two pools was significant (t = 3.96; *p* < 0.05). The AploB pDNA showed a more diverse salivary microbiota (H = 1.65) than the AploA pDNA (H = 1.56). At the genus level (Figure 4, Appendix A), the difference between diversity indices was higher (t = 11.73, *p* < 0.05), with the AploB pool showing a more diverse microbiota profile (H = 3.02) than the AploA pool (H = 2.66). The genera *Prevotella* and *Veillonella* were observed to be more abundant in the AploA pool, whereas only the genus *Porphyromonas* showed an appreciable increase in the AploB pool. At the species level (Appendix A), the results reflect those obtained at the genus and phylum level (Shannon H: AFRA = 3.70, AFRB = 3.97; t = 7.85, *p* < 0.05).

We noticed that the differences in taxa reflected those found between sexes. Accordingly, we repeated the analysis, dividing the pools into male Africans with diplotype aa (AploA-M), male Africans with diplotype bb (AploB-M), female Africans with diplotype aa (AploA-F), and female Africans with diplotype bb/ab (AploB-F). At the genus level (Appendix A), only comparisons between males showed a significant difference (Shannon H: AploAM = 2.82, AploBM = 3.12; t = 6.95, *p* < 0.05), contrary to females (Shannon H: AploAF = 2.51, AploBF = 2.54; t = 0.56, *p* > 0.05). Therefore, males with diplotype *TRPV1* bb possess the most diverse microbial community.

#### 2.2.4. Differences in Capsaicin Sensitivity

The comparison between the microbiota of individuals with different sensitivity to capsaicin was performed considering salivary pDNA of individuals with high sensitivity (HS, reads = 7630) and low sensitivity (LS, reads = 12,836).

At the phylum level (Appendix A), the diversity between the two pools was significant (Shannon H: t = 8.20; *p* << 0.05). The HS pool showed a more diverse salivary microbiota (H = 1.39) than the LS pool (H = 1.28). The more abundant phyla were *Proteobacteria* in the LS pool and Firmicutes and Actinobacteria in the HS pool. At the genus level (Figure 5, Appendix A), the two pools showed less different diversity values (t = 3.11, *p* < 0.05), with the HS microbiota slightly more diverse (Shannon H: HS = 2.69; LS = 2.62). The greater increase was for the genus *Escherichia* in the LS pool (+10.2%), for *Veillonella* and *Streptococcus* in the HS pool At the species level as well (Appendix A), the two groups differed significantly (t = 4.74; *p* < 0.05), with a higher microbiota diversity for the HS pool (Shannon H: HS = 3.75, LS = 3.65). *Haemophilus parainfluenzae* is the more abundant species in the HS pool (+6.9%) and *H. pitmaniae* and *E. coli* in the LS pool.

To investigate whether ethnicity influenced the diversity observed, HS and LS pDNAs were disaggregated according to ethnicity: Italians with high sensitivity (ITA-HS), Italians with low sensitivity (ITA-LS), Africans with high sensitivity (AFR-HS), and Africans with low sensitivity (AFR-LS). At the genus level (Appendix A), only comparisons between Italians showed a significant difference (Shannon H: ITA-HS = 2.75, ITA-LS = 2.61; t = 5.16, *p* < 0.05), not between Africans (AFR-HS = 2.46, AFR-LS = 2.50; 1.16, *p* > 0.05). To investigate whether sex affected diversity results, the pDNAs were divided into males with high (HS-M) and low (LS-M) sensitivity and females with high (HS-F) and low (LS-F) sensitivity. At the genus level (Appendix A), only male comparisons gave a significant difference (Shannon H: HS-M = 2.51, LS-M = 2.41; t = 2.67, *p* << 0.05), not female comparisons (Shannon H: HS-F = 2.61, LS-F = 2.59; t = 0.77, *p* > 0.05). Therefore, individuals with high sensitivity showed a more diversified salivary microbiota. This finding is emphasized according to sex (males) and ethnicity (Italians).

## 3. Discussion

Saliva is considered a reservoir and even a fingerprint of the entire oral microbiota, although not representative of the entire population of oral districts. 

Despite critical points such as unintentional sampling, suboptimal DNA quality (stored at −20 °C for 4 years), and the fact that oral microbiota studies are affected by the different sequencing approaches used [42], our results seem to suggest a satisfactory reliability of the methodological approach. Compared to Illumina-based metagenomics, the long reads produced by the ONT approach allow sequences to be easily assigned to an operational taxonomic unit (OTU), even at the species and strain level. In contrast, ONT sequencing requires higher purity and quantity of DNA (>400 ng DNA) to ensure successful sequencing and suffers from a higher error rate (around 5% with R9.4 chemistry). The number of species assigned at different sequencing depths (rarefaction curves, Appendix A) reached a plateau for each aggregated pDNAs, ensuring an optimal coverage of microbial composition. In addition, the overall results agree with the literature, confirming that the oral microbiota is formed by a more stable core and a more variable spectrum of taxa. The variations between the different groups concerned mainly changes in relative abundances of taxa, rather than changes in the presence/absence of single microorganisms. Therefore, our results can be considered a new contribution to understanding the variations in composition of the human oral microbiota.

We found significant variations in salivary microbial diversity based on ethnicity, sex, genetic diversity (*TRPV1* diplotypes), and chemesthetic perception (sensitivity to capsaicin). Sub-Saharan Africans showed a more diversified and richer microbiota than the Italians, with a greater abundance of the genera *Streptococcus* and *Veillonella* (Firmicutes) and *Rothia* (Actinobacteria). This finding agrees with the studies of Yang et al. [43] and Li et al. [11], in which Africans showed a more diverse microbiota than Germans, native Alaskans and native Americans. Furthermore, our data confirm that ethnicity exerts selective pressure on the oral microbiota, as previously stated by Mason and colleagues [44].

Several studies have shown how a habitual diet can determine changes in the composition of the intestinal microbiota, particularly the Mediterranean one, characterized by intake of cereals, fruits, and vegetables in greater quantities than other European diets [45,46]. It is therefore possible to hypothesize a substantial influence of the nutrients typical of the Mediterranean diet (vitamins B and E), polyphenols, and other bioactive compounds on the composition of the oral microbiota.

The comparison between sexes reported an increase in the genera *Rothia*, *Veillonella,* and *Neisseria* in the females compared to males, despite the latter showing a richer and more diversified microbiota. To date, there are few studies about this comparison. However, in the work of Murugesan and collaborators [13], on the population of Qatar, such evaluation did not recognize a significant difference in the two groups, even if the females were more abundant for *Treponema* and *Mycoplasma*. In a more recent study [47], distinct differences in the predominant microbial genera between females and males were found. For females, the most abundant genera included *Streptococcus*, followed by *Prevotella* and *Granulicatella*. In contrast, the saliva of males showed a different profile, with *Campylobacter* A being the most prevalent, followed by *Veillonella*, *Porphyromonas*, and *Oribacterium*. The significant differences in salivary microbiota composition between the sexes indicate that gender plays a crucial role in defining the microbial diversity and abundance in the mouth.

This is one of the first studies investigating the influence of genetic variability on the oral microbiota. We demonstrated that individuals with haplotype *TRPV1* H1-b, in the homozygous or heterozygous state, were richer in taxa than the carriers of the haplotype *TRPV1* H1-a. In the study of Giannì and collaborators [41], a correlation between *TRPV1* diplotypes and body composition of adaptive type was hypothesized, consisting of responses controlled by different diplotypes in response to heterogeneous environmental conditions. In African women carrying the H1-b haplotype, a higher percentage of fat mass and lower extracellular fluid retention were observed. This suggests the possible action of sex-driven balancing selection at the non-coding sequences of the *TRPV1* gene, with advantageous adaptive effects for people living in arid areas with strong daily or seasonal temperatures. Accordingly, the greater microbiota diversity of the H1-b haplotype carriers might be related to the more intense environmental and food fluctuations. More targeted studies are needed to verify this hypothesis. 

As far as we know, this is the first study that correlated capsaicin sensitivity with salivary microbiota. The scientific community has focused the research on the link between capsaicin and gut microbiota. Therefore, although we are aware that there is often a close relationship, we could not make a direct comparison of our findings with literature. We can hypothesize that a lower sensitivity to capsaicin (possibly associated with a higher consumption of chili peppers and other hot food) may correspond to a reduction in the diversification of the oral microbiota. In fact, capsaicin, depending on its concentration and on bacterial strain, has been demonstrated to exert an inhibitory effect against several bacteria [48]. Furthermore, according to Menicagli, et al. [49], the intake of capsaicin leads to an increase in oxidative stress within the oral cavity, with increased production of malondialdehyde and later salivary nitric oxide) conditions involving physiological changes in the oral cavity. This change in oral “environmental” conditions could lead to an alteration of the microbiota in terms of a decrease in diversity and the spread of species with pathogenic potential (*Escherichia coli* + 11%, *Haemophilus pittmaniae* + 6.4%).

## 4. Materials and Methods

### 4.1. Sample Selection

The recruited samples (Appendix A) were the same as in Giannì and colleagues (2024) [41]: 46 African healthy donors (AFR) (34 males and 12 females) originating in various sub-Saharan countries and 45 Italians (21 males and 24 females) (ITA). By signing a consent form in accordance with the current regulations on the processing of personal data, volunteers agreed to perform a sensory sensitivity test for capsaicin and donate a saliva sample for genetic analysis.

All extracted DNA samples were subjected to spectrophotometric analysis to assess their concentration (260/280 nm ratio) and purity (260/230 nm ratio). DNA samples showing the highest values were selected. Two regions of the *TRPV1* gene were sequenced by the cycle sequencing method as previously reported [41].

### 4.2. Rapid Barcoding Sequencing Kit (SQK-RBK004) and DNA Pool (pDNA)

Metagenomic analyses were performed by Nanopore sequencing technology (ONT: Oxford Nanopore Technologies) on the MinION device. Pools of DNA (pDNA) were created using DNA from different samples. Each pDNA consisted of a mixture of equal amounts of DNA per sample for a total of 400 ng. 

To investigate differences based on ancestry and sex, we selected pDNAs with 16 Africans (8 males and 8 females) and 12 Italians (6 males and 6 females). To investigate differences based on genetic variants, we selected pDNAs with bb/ab *TRPV1* diplotypes (8 Africans) and aa *TRPV1* diplotypes (8 Africans). To investigate differences based on sensitivity to capsaicin, we selected pDNAs from individuals at the extremes of a sensitivity scale from 1 to 10: volunteers with a 1–3 sensitivity (LS, 10 Africans and 10 Italians) and 6–9 sensitivity (HS, 10 Africans and 10 Italians) [41].

The kit SQK-RBK004 was used to label each pDNA for simultaneous sequencing runs. To do this, each pDNA was assigned a specific barcode, via the barcode trasposome complex, which reduces the DNA into pair-end fragments and labels them with specific oligos. To each library was added the rapid adapters (RAP) and a motor protein to guide the individual fragments through the nanopores (flow cell R9.4.1). 

### 4.3. Sequencing and Basecalling

Sequencing runs were set by the software MinKNOW version 20.10 (Oxford Nanopore Technology). It allowed us to perform the hardware and flow cell check and to set the running parameters, such as duration (72 h), voltage power (180 mV), input and output folder, and minimum signal quality (7). The base-calling (fast accuracy mode) was performed through the Guppy software, and the reads were saved in fastQ files.

### 4.4. Taxonomic Assignment

To obtain the taxonomic assignment of the reads that passed quality control (passed), the EPI2ME software package was used. It allowed an end-to-end analysis of fastQ files via a cloud. For this study, we used the algorithm WIMP (what’s in my pot?). WIMP uses the KRAKEN algorithm, which can assign taxonomic labels to DNA sequences through the use of k-mer with a speed higher than BLAST but with comparable accuracy. The algorithm works using a pre-constructed structure that maps oligonucleotides (k-mers) with a length of 24 bp at the taxonomic nodes of the NCBI reference tree. A taxon is assigned to the reads when, at the level of the individual nodes, the respective 24 mer diagnostics are recognized. Thanks to this procedure, the assignment is faster, as complete alignment to the original reference sequence is not required. The final output is a report containing a distribution of reads/taxon at different taxonomic levels (phylum, family, genus, species) and a tree of OTUs assigned to different levels of relative abundance of reads [50]. Once the reads were classified, they were subjected to a post-run quality check, consisting of the elimination of reads classified as *Homo sapiens*, “root”, “unclassified”, and those that showed an uncertain taxonomy. The reads that had only one occurrence were subsequently eliminated to avoid stochastic bias in index determination and statistical testing.

Relative abundances of taxa were calculated by simply dividing the occurrence value of the taxon in question by the total of the reads. The reads were then subjected to the rarefaction process before performing statistical analyses (Appendix A).

### 4.5. Statistical Analyses

#### 4.5.1. Alpha Diversity

In ecology, the term alpha diversity refers to the diversity within a single community or sample, and it allows describing the structure of an ecological community in terms of richness in species (the number of taxonomic groups detected within the sample). Another type of metric is represented by so-called diversity indices that allow to measure the distribution of taxa within the community.

For this project, as alpha metrics, we used the richness (S) and the Shannon diversity index (H). Richness is the number of taxa in the sample compared to the total number of sequenced reads. The Shannon diversity index allows us to describe how evenly the taxa are distributed within the sample. This index is one of the most widely used metrics to quantify the composition of an ecosystem, as both rare and common species are considered [51]. To perform pools rarefaction and calculate the different indices of alpha diversity (S and H), the vegan package of R was used, respectively, through the functions “rarefy” and “diversity”.

#### 4.5.2. Hutcheson *t*-Test

We used the Hutcheson *t*-test, instead of the ordinary student *t*-test, to compare the differences between two pDNAs in the absence of repeated observations. This is a non-parametric method that allows to compare two samples using only the values of the diversity index (Shannon H) and their variance. Thus, for communities with abundance in the order of hundreds or thousands of organisms, the critical value of significance (0.05) is about 1.96 [52]. The R packages “ecoltest” and “Hutchetson *t*-test” were used.

## 5. Conclusions

The study is one of the first to investigate the influence of genetic variability on salivary microbiota composition and provides insights into its diversity in relation to endogenous factors such as ethnicity, sex, and genetic variants of the *TRPV1* gene. It also explores the relationship between capsaicin sensitivity and salivary microbiota composition. The results indicate that while the overall structure of the salivary microbiota remains stable, variations in the relative abundances of specific taxa are observed among different ethnic groups. This suggests that individual characteristics, both due to genetics and culturally related dietary practices, may play a crucial role in shaping the salivary microbiota. Our results may form the basis for subsequent studies in which other ethnic groups and genes could be considered.

## Figures and Tables

**Figure 1 ijms-25-11585-f001:**
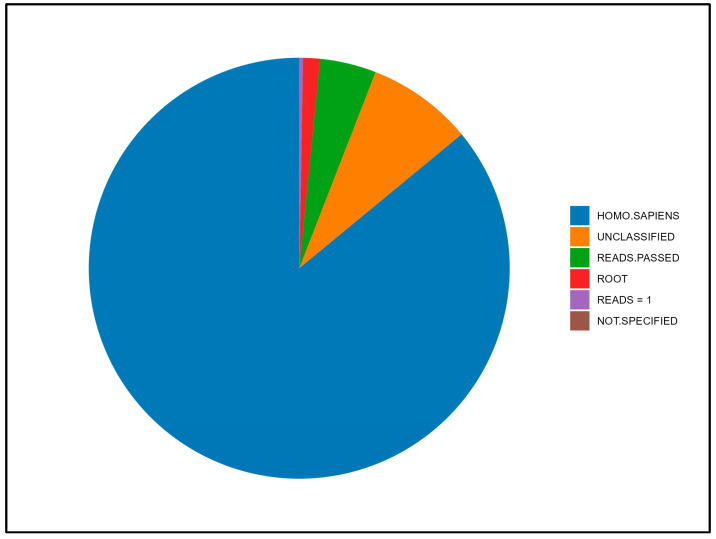
Reads subjected to post-run quality control.

**Figure 2 ijms-25-11585-f002:**
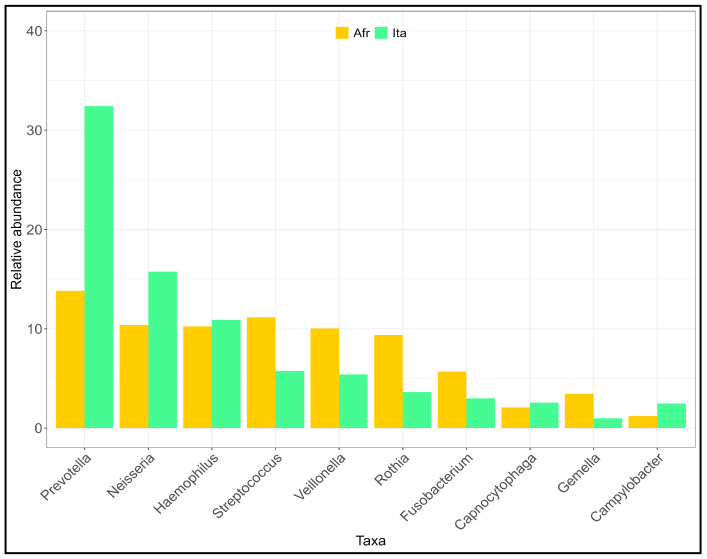
Distribution of the more abundant genera in sub-Saharan Africans (AFR) and Italians (ITA). The Hutcheson test returned a significant difference between the two groups at phylum (t = 9.73, *p* < 0.001), genus (t = 13.46; *p* < 0.001) and species (t = 11.32; *p* < 0.001) level.

**Figure 3 ijms-25-11585-f003:**
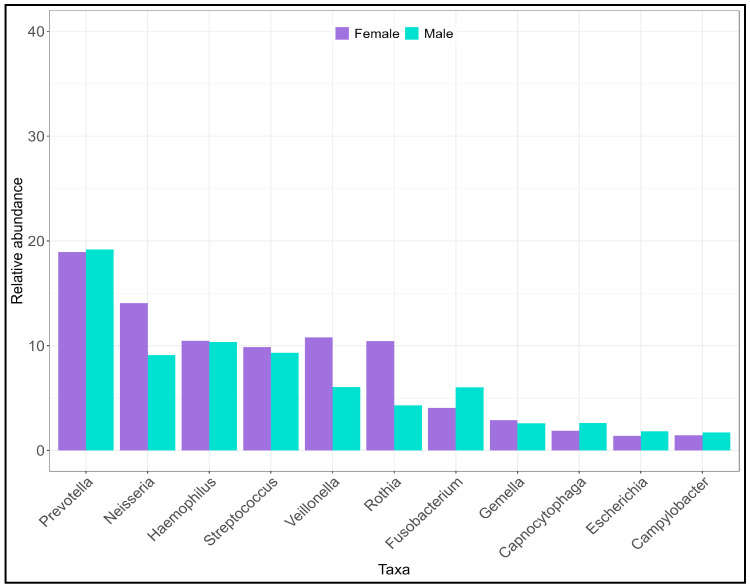
Distribution of the more abundant genera in males (M) and females (F). The Hutcheson test returned a significant difference between the two groups at phylum (t = 3.11, *p* < 0.05), genus (t = 13.71; *p* < 0.001) and species (t = 11.6; *p* < 0.001) level.

**Figure 4 ijms-25-11585-f004:**
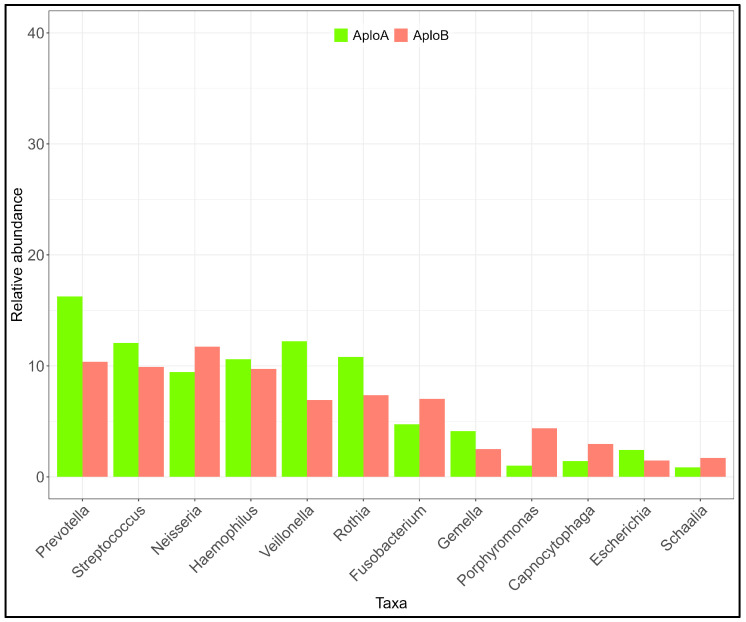
Distribution of the more abundant genera in the *TRPV1* aa (AploA) and bb/ab (AploB) diplotype pools. The Hutcheson test returned a significant difference between the two groups at phylum (t = 3.96, *p* < 0.001), genus (t = 11.73; *p* < 0.001 and species (t = 7.85; *p* < 0.001) level.

**Figure 5 ijms-25-11585-f005:**
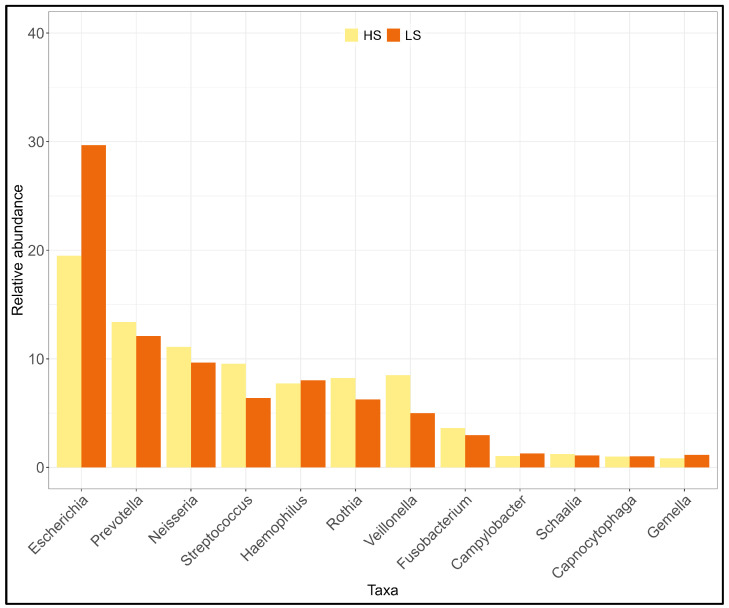
Distribution of the more abundant genera in DNA pools with different sensitivity to capsaicin: high sensitivity (HS) and low sensitivity (LS). The Hutcheson test returned a significant difference between the two groups at phylum (t = 8.20, *p* < 0.001), genus (t = 3.11; *p* < 0.05) and species (t = 4.74; *p* < 0.001) level.

## Data Availability

The original contributions presented in the study are included in the article/Appendix A; further inquiries can be directed to the corresponding author.

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
