# Peer review of "Human Salivary Microbiota Diversity According to Ethnicity, Sex, TRPV1 Variants and Sensitivity to Capsaicin"

_ijms, 2024, doi:10.3390/ijms252111585_

Round 1

Reviewer 1 Report

Comments and Suggestions for Authors

This manuscript addresses different characteristics in bacterial richness and diversity in the oral microbiota, in two populations with different ethnicity, different genera within the different ethnicities. In addition, they present different interesting parameters but in a way that is in my opinion not very orderly, in some cases only in one of the cohorts (African). .
The manuscript that has to do according to its title with Human salivary microbiota diversity according to ethnicity, sex, TRPV1 variants and sensitivity to capsaicin uses an unusual methodology in this type of analysis, performing a pool of saliva from the two populations with different ethnicity, the figures that accompany the text do not show the dispersion of the data. I think that the sensible thing would have been to perform the sequencing of at least the (16S) gene for the fairly simple rRNA. On the other hand, the supplementary material is not tables but rather Excel spreadsheets without any kind of treatment, although I must say that even though the information is there, it is not easy to analyze it.
The sampling provided much more than the information shown. In addition, there is no paragraph regarding the possible limitations of its methodological strategy and no section of conclusions on the results obtained.

Author Response

We sincerely thank the reviewer for his/her comments and suggestions. We reply point -by-point:

Comments 1: …they present different interesting parameters but in a way that is in my opinion not very orderly, in some cases only in one of the cohorts (African)

Response 1: We have presented the diversity found according to the order indicated in the title: ‘by ethnicity, by sex, by TRPV1 variants and by capsaicin sensitivity’. Within each section, we described the abundance of the microbiota according to a descending taxonomic order, from Phylum to Species level. The comparison concerning the composition of the microbiota according to TRPV1 haplotypes was performed only in the African cohort because it was the only population sample with both haplotypes (a and b).

Comments 2: …the figures that accompany the text do not show the dispersion of the data

Response 2: The figures do represent the dispersion of the data because they show the presence of the OTUs making up the microbiotic communities in the form of relative abundance. For obvious reasons of space and readability, we have shown only the major differences, referring full distributions to supplementary tables.

Comments 3: …I think that the sensible thing would have been to perform the sequencing of at least the (16S) gene for the fairly simple rRNA

Response 3: The use of long reads and the KRAKEN algorithm made it possible to assign a taxonomic label using a wide range of genomic sequences beyond 16S, while maintaining an accuracy comparable to that of a BLAST search.

Comments 4: …the supplementary material is not tables but rather Excel spreadsheets without any kind of treatment, although I must say that even though the information is there, it is not easy to analyze it.

Response 4: We followed the reviewer's suggestion and converted the Excel spreadsheets into pdf tables. We have also added a legend to each table for easier interpretation.

Comments 5: …there is no paragraph regarding the possible limitations of its methodological strategy and no section of conclusions on the results obtained

Response 5: In the discussion section, we added a paragraph concerning the possible limitations of the methodological approach we followed:

‘Compared to Illumina-based metagenomics, the long reads produced by the ONT approach allow sequences to be easily assigned to an Operational Taxonomic Unit (OTU), even at the species and strain level. In contrast, ONT sequencing requires higher purity and quantity of DNA (>400 ng DNA) to ensure successful sequencing and suffers from a higher error rate (around 5% with R9.4 chemistry). The number of species assigned at different sequencing depths (rarefaction curves, Figure S1) reached a plateau for each aggregated pDNAs, ensuring an optimal coverage of microbial composition.’[This change can be found at page 7, line 234]

Conclusions are not mandatory in the format of the article, and the discussion already summarises well the results, which are complex and therefore cannot be reduced to a few lines. In any case, we gladly accept the reviewer's suggestion and add the following conclusion:

‘Our study is one of the first to investigate the influence of genetic variability on the composition of the salivary microbiota and provides insights into its diversity in relation to endogenous factors such as ethnicity, gender and genetic variants of the TRPV1 gene. It also explores the relationship between capsaicin sensitivity and salivary microbiota composition. The results indicate that while the overall structure of the salivary microbiota remains stable, variations in the relative abundances of specific taxa are observed between different ethnic groups. This suggests that individual characteristics, due to both genetics and culturally related dietary practices, may play a crucial role in shaping the salivary microbiota. Our results may form the basis for subsequent studies in which other ethnic groups and genes may be considered'. [This change can be found at page 11, line 402]

Reviewer 2 Report

Comments and Suggestions for Authors

he manuscript explores the salivary microbiota of two distinct ethnic groups: Italians and sub-Saharan Africans. The authors examine both endogenous and exogenous factors that influence variations in microbiota diversity. While the manuscript is well-written and clearly presented, I have a few suggestions that may enhance its quality:

  • The authors included two ethnic groups, Italian and African. While these populations have notable differences, it might be beneficial to include other ethnicities, such as those from colder regions, to see if environmental temperature impacts microbiota composition. Studies like Geng et al. (2023), which looks at how temperature and altitude affect the gut microbiota of athletes, and Wang et al. (2024), which discusses how gut microbiota help hosts adapt to extreme temperatures, could provide valuable context.

  • Why did the authors focus solely on TRPV1 and not explore other members of the TRP channel family? Do they have any data on additional members of this extensive family?

  • The figures lack indications of statistical significance. Including these would be useful.

  • A summary table presenting information about the populations would also be helpful for readers.

Author Response

We sincerely thank the reviewer for his/her comments and suggestions. We reply point -by-point:

 Comments 1: The authors included two ethnic groups, Italian and African. While these populations have notable differences, it might be beneficial to include other ethnicities, such as those from colder regions, to see if environmental temperature impacts microbiota composition. Studies like Geng et al. (2023), which looks at how temperature and altitude affect the gut microbiota of athletes, and Wang et al. (2024), which discusses how gut microbiota help hosts adapt to extreme temperatures, could provide valuable context.

Response 1: We are aware that it would be useful to add other ethnic groups to the analysis. However, this work relates to a previous study that highlighted two haplotype blocks in the TRPV1 gene (identified by an extensive in silico analysis of previous genetic studies deposited in global databases) with strong stabilising selection signals only in individuals of sub-Saharan African descent [Giannì, M.; Antinucci, M.; Bertoncini, S.; Taglioli, L.; Giuliani, C.; Luiselli, D.; Risso, D.; Marini, E.; Morini, G.; Tofanelli, S. Association between Variants of the TRPV1 Gene and Body Composition in Sub-Saharan Africans. Genes (Basel) 2024, 15, 752, doi:10.3390/genes15060752]. For this reason, variants from these two regions have been analysed in volunteers of sub-Saharan African descent and in Italian volunteers of both sexes.

Comments 2: Why did the authors focus solely on TRPV1 and not explore other members of the TRP channel family? Do they have any data on additional members of this extensive family?

Response 2: While several works exist on the influence of capsaicin (which activates TRPV1) on the composition of the gut microbiota, only a few have studied other genes of the TRP channels family. Importantly, our work is one of the first to investigate the influence of genetic variability on the salivary microbiota and these early data may be the basis for further studies in which, as suggested, other ethnic groups (as well as other genes) could be considered.

Comments 3: The figures lack indications of statistical significance. Including these would be useful.

Response 3: We have added statistical significance in the figure legend and in the text. We hope this will help readers assess the differences between the pDNA distributions.

 Comments 4: A summary table presenting information about the populations would also be helpful for readers.

Response 4: We accepted the reviewer's suggestion and included a synoptic table of sample compositions as supplementary Table S1.

Round 2

Reviewer 1 Report

Comments and Suggestions for Authors

The present  revised version of the manuscript  was substantially improved, however, I have personally a conflict  with sample pools in this type of research ,  but reviewing the related literature, I found some articles that report results of such  approach strategy.